# Study on the Preparation Technology of a Ceramic Panel with a Magnetic Interlayer for an Induction Cooker

**Aijin Pan [1], Haifeng Lan [1], Yongjun Huang [2], Peng Chen [1] and Shuangxi Wang [1,*]**

[1] College of Engineering, Shantou University, Shantou 515063, China; 17ajpan@stu.edu.cn (A.P.); 16hflan@stu.edu.cn (H.L.); 17pchen3@stu.edu.cn (P.C.)

[2] Chaozhou Three-Circle (Group) Co., Ltd., Chaozhou 521000, China; 15yjhuang3@alumni.stu.edu.cn

[*] Correspondence: sxwang@stu.edu.cn; Tel.: +86-0754-8251-8903

**Abstract:** In order to expand the range of pot materials for induction cookers, a kind of sandwich structural composite ceramic panel that consists of an $Al_2O_3$ ceramic substrate, magnetic heating interlayer, and $ZrO_2$ ceramic substrate was developed by combining the tape casting process and the screen printing process. In this research, the slurry composition of the functional phase, glass powder, and organic carrier was optimized for preparing the heating interlayer with excellent electromagnetic properties. The influences of the glass powder content and the magnetic layer structure on the thermal shock resistance of the composite ceramic panel were studied. The finite element model of the composite ceramic panel under thermal load was established through ANSYS software. In the range of 0.1–0.3 mm thickness of a magnetic heating interlayer, the temperature field and the macroscopic stress field of the composite ceramic panel were simulated, and the influence of the magnetic layer structure on the thermal stress distribution of the composite ceramic panel was analyzed. The experimental results showed that the magnetic layer had the best quality when the amount of glass powder added was 9 wt%. The ANSYS simulation revealed that the gradient structure of the magnetic layer could reduce the stress between the alumina layer and the magnetic layer from 308 to 192 MPa, which significantly improved the thermal shock resistance of the composite ceramic panel. Therefore, the gradient structure of the magnetic layer could ensure the stability of the composite ceramic panel after five cycles of electromagnetic heating.

**Keywords:** induction heating; composite ceramic panel; screen printing; electromagnetic slurry; finite element analysis

## 1. Introduction

Ceramics have been used as a material in pots for many years. Nowadays, China has the world's largest ceramic pot production base. Compared to metal pots, pots made of ceramic have a lower cost, are wear resistant, corrosion resistant, and can maintain the original taste of food, which is why it is considered to be one of the healthiest and safest pot materials. Meanwhile, it is also environmentally friendly [1].

Compared with other heating techniques such as flame heating and resistance heating, induction heating technology has been widely used due to its advantages of fast heating, high thermal efficiency, cleanliness, safety, and accurate control [2–4]. In recent years, with the introduction of new national laws and regulations on energy and environmental protection, and an increase in people's awareness of energy conservation and environmental protection, induction heating technology has made great progress in the kitchen appliance industry [5,6].

However, the traditional ceramic pot is difficult to apply in induction heating due to its non-magnetic and non-conducting characteristics. If a ceramic pot could be combined with electromagnetic induction technology, not only could it improve the heating efficiency of the ceramic pot but it could also maintain the taste of the food itself, which conforms to modern people's dietary concept of health, nutrition, and environmental protection. Therefore, in 2006, the Japan Tiger Magic Bottle Co., Ltd., sprayed a far-infrared ceramic coating on the surface of a copper composite container so that the electromagnetic rice cooker took into account the advantages of thermal efficiency and uniform heat transfer and the slow heat dissipation of a traditional casserole [7]. In 2015, Tiger introduced a full-clay electromagnetic rice cooker, which was formed by spraying a layer of a magnetic heater on the bottom of the full-clay earth cooker and firing it at high temperature [8]. By embedding a ferromagnetic plate in the bottom of a sintered ceramic pot, Chen [9] succeeded in heating a ceramic pot by electromagnetic induction. Although the existing technology has achieved the purpose of electromagnetic heating by adding ferrous metal materials to ceramic pots by pasting, electroplating, and inlays, this technique is liable to cause the cracking of ordinary ceramic pots due to the difference in the thermal expansion coefficients between the metal materials and the ceramic. Moreover, when the ceramic pot is washed and moved, it will also cause wear and tear to the bottom metal material.

The sandwich structure—which consists of two external high-density materials and a thick core made from low-density material—has been widely applied in aerospace, the aircraft and marine industries, and in civil engineering. The configuration coupled with an optimized material combination significantly improves the overall performance of the system and structure, as well as the efficiency of material use. However, the working environment of high temperature and impact load will cause problems such as cracks in the sandwich structure. Based on a moving mesh approach, Marco et al. [10] proposed a numerical method methodology to investigate the behavior of composite sandwich structures under static and dynamic loading conditions. It could correctly simulate interfacial crack onset, layer kinematic, and debonding propagation. Based on 3D FEM (Finite Element Method), Xue et al. [11] simulated the thermal-structural response of complicated sandwich composites. FEM has been one of the most effective software for assisting manufacture.

In this paper, a composite ceramic panel with a sandwich structure was prepared. In Section 2, the finite element model of alumina–magnetic material–zirconia (in a sandwich structure) composite ceramic panel under thermal load was established by ANSYS software. The temperature field and the macroscopic stress field of the composite ceramic panel were simulated. In Section 3, based on the results of the finite element analysis, a magnetic slurry was developed, and screen printed onto the tape casted $Al_2O_3$ and $ZrO_2$ ceramic panel. The composite ceramic panel with a gradient magnetic interlayer was sintered at 950 °C under a protective atmosphere. Finally, in Section 4, the influence of the magnetic layer structure on the thermal stress distribution of the composite ceramic panel was analyzed. The micromorphology and thermal shock resistance behavior of the panel under different conditions were investigated.

## 2. Modeling of the Sandwich Structure Composite Ceramic Panel

### 2.1. Structure Model of the Composite Ceramic Panel

The temperature field and stress field of the alumina–magnetic material–zirconia (sandwich structure) model heated at 900 W for 10 s on an electromagnetic furnace were analyzed. Figure 1 shows the structure model of the composite ceramic panel. $\delta_1$ is the thickness of the alumina layer; $\delta_2$, the magnetic layer; and $\delta_3$, the zirconia layer. According to the formula of the magnetic slurry in Table 1, the A3 sized sample was selected as a single layer structure $\delta_2$ in Figure 1a, and A5, A3, and A4 sized samples were selected for the three-layer gradient structure of $\delta_{21}$, $\delta_{22}$, and $\delta_{23}$ in Figure 1b, respectively. To simplify the calculation and analysis, the stress and strain of the model with a section length of 5 mm during heating were analyzed by ANSYS software.

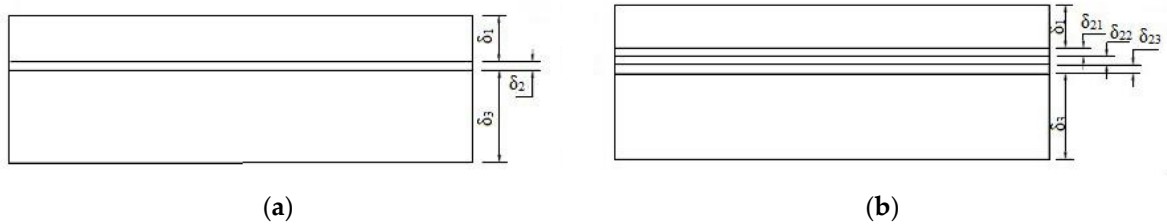

**Figure 1.** Model of the composite ceramic panel (**a**) with a single magnetic interlayer and (**b**) with a three-layer gradient.

**Table 1.** Constituents of the magnetic slurry (wt%).

| No. | Copper Powder | Iron Powder | Glass Powder | Organic Carrier |
|---|---|---|---|---|
| A1 | 62 | 20 | 3 | 15 |
| A2 | 59 | 20 | 6 | 15 |
| A3 | 56 | 20 | 9 | 15 |
| A4 | 53 | 20 | 12 | 15 |
| A5 | 50 | 20 | 15 | 15 |

### 2.2. Geometric Characteristics and Mesh Generation of the Model

The model parameters are shown in Tables 2 and 3. When the intermediate magnetic layer is a three-layer gradient structure, the relationship between the layers presents a gradient change. The gradient layer is regarded as a series of well-bonded composite layers, and each layer has different material properties. In the calculation process, it is assumed that the thickness of the upper and lower layers of the model remains constant and that the thickness of the middle layer is a variable.

**Table 2.** Main parameters of the model with a single magnetic interlayer.

| Parameter | Numerical Value (mm) |
|---|---|
| Thickness of Alumina layer $\delta_1$ | 0.5 |
| Thickness of magnetic layer $\delta_2$ | 0.1, 0.2, 0.3 |
| Thickness of Zirconia layer $\delta_3$ | 1.0 |
| Side length of composite ceramic panel | 5.0 |

**Table 3.** Main parameters of the model with a three-layer gradient.

| Parameter | Numerical Value (mm) |
|---|---|
| Thickness of Alumina layer $\delta_1$ | 0.5 |
| Thickness of magnetic layer $\delta_{21}$ | 0.1 |
| Thickness of magnetic layer $\delta_{22}$ | 0.1 |
| Thickness of magnetic layer $\delta_{23}$ | 0.1 |
| Thickness of Zirconia layer $\delta_3$ | 1.0 |
| Side length of composite ceramic panel | 5.0 |

In the ANSYS analysis, due to the geometric symmetry of the sample, a two-dimensional planar model was used to simplify the calculation of the model. The finite element meshes after modeling are shown in Figures 2 and 3. By dividing the continuum into a limited number of elements, then combining these elements into a whole, introducing boundary conditions, and establishing algebraic equations, the displacement and stress of the continuum at discrete points are finally obtained. The unit length of mesh generation was 0.05 mm, and the X and Y direction constraints were applied in the lower left corner.

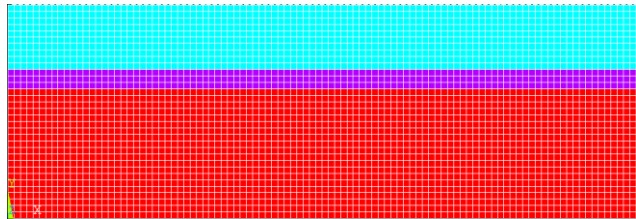

**Figure 2.** Finite element mesh of the specimen with the single magnetic interlayer.

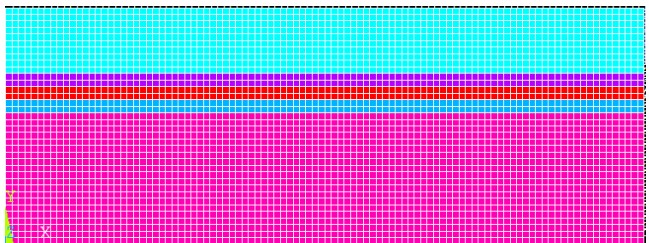

**Figure 3.** Finite element mesh of the specimen with the three-layer gradient.

*2.3. Description of the Properties of the Magnetic Layer Materials*

In the study of composite properties, the modified mixing law proposed by Tamura et al. is the most commonly used method for calculating material parameters [12]. In this paper, the properties of each layer in the three-layer magnetic gradient layer were different, and the gradient layer with the different volume fraction of components was simulated according to the multi-layer structure [13]. In this experiment, the magnetic layer contained three phases: copper powder, iron powder, and glass powder, and the Poisson's ratio μ of each composite of the magnetic layer was calculated according to Equation (1):

$$\mu = V_A \mu_A + V_B \mu_B + V_C \mu_C \tag{1}$$

where $\mu_A$, $\mu_B$, and $\mu_C$ are the Poisson's ratio of the copper powder, iron powder, and glass powder, respectively, and $V_A$, $V_B$, and $V_C$ are the volume fraction of the copper powder, iron powder, and glass powder in the magnetic layer, respectively.

The thermal expansion coefficient λ of the magnetic layer was calculated by the double-layer plate model proposed by Gulati [14], which is given by:

$$\lambda = (\lambda_A V_A K_A + \lambda_B V_B K_B + \lambda_C V_C K_C)^{-1} \tag{2}$$

$$k = \frac{E}{2(1-\mu)} \tag{3}$$

where $E$ is the modulus of elasticity; $\lambda_A$, $\lambda_B$, and $\lambda_C$ are the thermal expansion coefficients of the copper powder, iron powder, and glass powder, respectively.

The thermal conductivity of the magnetic layer was calculated according to the thermal conductivity model (Equation (4)) of the three-phase system given in [15]:

$$k_e = \frac{k_A V_A \frac{2k_A + k_C}{3k_A} + k_B V_B \frac{2k_B + k_C}{3k_B} + k_C V_C}{V_A \frac{2k_A + k_C}{3k_A} + V_B \frac{2k_B + k_C}{3k_B} + V_C} \tag{4}$$

where $k_e$ is the effective thermal conductivity of the magnetic layer, and $k_A$, $k_B$, and $k_C$ are the thermal conductivities of the copper powder, iron powder, and glass powder, respectively.

The performance parameters of each material in the model are given in Tables 4 and 5. In order to simplify the calculation, the effect of temperature on the performance parameters of the alumina and zirconia substrates during electromagnetic heating was neglected.

**Table 4.** Material parameters of the model with a single layer.

| Material | Density/ gcm$^{-3}$ | Coefficient of Thermal Expansion/ ppm·K$^{-1}$ | Thermal Conductivity/ W·(m·K)$^{-1}$ | Modulus of Elasticity/ Em·GPa$^{-1}$ | Poisson Ratio | Specific Heat Capacity |
|---|---|---|---|---|---|---|
| Alumina layer | 3.97 | 8.80 | 23.0 | 390 | 0.240 | 750 |
| Magnetic layer | 6.74 | 15.05 | 218.2 | 114 | 0.285 | 477 |
| Zirconia layer | 6.00 | 10.80 | 1.2 | 210 | 0.300 | 710 |

**Table 5.** Material parameters of the model with a gradient structure.

| Material | Density/ g·cm$^{-3}$ | Coefficient of Thermal Expansion/ ppm·K$^{-1}$ | Thermal Conductivity/ W·(m·K)$^{-1}$ | Modulus of Elasticity/ Em·GPa$^{-1}$ | Poisson Ratio | Specific Heat Capacity |
|---|---|---|---|---|---|---|
| Alumina layer | 3.97 | 8.80 | 23.0 | 390 | 0.240 | 750 |
| Magnetic layer 1 | 5.46 | 13.46 | 127.9 | 95 | 0.272 | 567 |
| Magnetic layer 2 | 6.74 | 15.05 | 218.2 | 114 | 0.285 | 477 |
| Magnetic layer 3 | 5.97 | 14.25 | 167.9 | 103 | 0.277 | 532 |
| Zirconia layer | 6.00 | 10.80 | 1.2 | 210 | 0.300 | 710 |

## 3. Experiment

### 3.1. Preparation and Coating of the Magnetic Slurry

The composition of the magnetic slurry is given in Table 1. In Table 1, the particle median diameter of copper powder, iron powder, and glass powder are 1, 1, and 3 μm, respectively, and the information on the composition of the glass powder and organic carrier are given in Tables 6 and 7. The slurry was homogenized by milling for 20 to 60 min. Then, the slurry was screen printed onto the upper surface of the self-made $ZrO_2$ ceramic plate with a thickness of 1 mm and a size of 100 × 100 mm and printed on the lower surface of the self-made $Al_2O_3$ ceramic panel with a thickness of 0.5 mm and a size of 100 × 100 mm. The thickness of the coating obtained was about 100 μm. Next, the lower surface of the $Al_2O_3$ ceramic panel was closely attached to the upper surface of the $ZrO_2$ ceramic substrate. After drying at 100 °C for 15 min, samples with the slurry coating were prepared.

**Table 6.** Composition of the organic carrier.

| Material | Terpineol | Tributyl Citrate | Ethyl Cellulose | Lecithin | Hydrogenated Castor Oil |
|---|---|---|---|---|---|
| Content (wt%) | 60 | 23 | 6 | 5 | 6 |

**Table 7.** Composition of the glass powder.

| Material | $Bi_2O_3$ | $B_2O_3$ | ZnO | $SiO_2$ |
|---|---|---|---|---|
| Content (wt%) | 45 | 20 | 8 | 27 |

The fabrication process of the composite ceramic panel with a single magnetic interlayer is shown in Figure 4. Considering the difference in thermal expansion coefficients between zirconia and alumina ceramics, A4, A3, and A5 slurries were sequentially printed onto the upper surface of the $ZrO_2$ ceramic substrate, and an A5 slurry was printed onto the lower surface of the $Al_2O_3$ ceramic panel. After drying and laminating, the composite ceramic panel with a magnetic gradient coating was obtained.

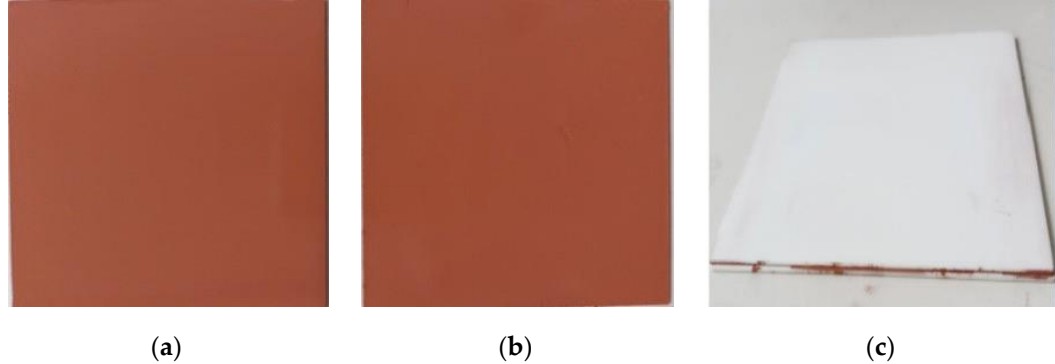

**Figure 4.** Fabrication process of the composite ceramic panel with a single magnetic interlayer: (**a**) alumina substrate with a magnetic slurry; (**b**) zirconia substrate with a magnetic slurry; (**c**) composite ceramic panel with a single magnetic interlayer.

### 3.2. Sintering of the Composite Ceramic Panel

The composite ceramic panels with the magnetic coating were fired in an atmosphere furnace by a four-step heating process: first, the temperature was rapidly increased from room temperature to 800 °C at 20 °C/min; second, the temperature was increased to 950 °C at 15 °C/min; third, the temperature was maintained for 30 min; and fourth, the temperature was cooled to room temperature naturally. Alumina has good thermal conductivity and zirconia has a good thermal barrier. The composite ceramic panel with this structure not only ensures its excellent mechanical properties but also transfers the heat generated by the electromagnetic interlayer to the cooking appliance as much as possible.

### 3.3. Measurement and Characterization

The thermal expansion coefficients of the coatings were measured using a thermal dilatometer (NETZSCH Instruments, DIL402C model, Selb, Germany). The surface and cross-section morphology of the magnetic layer of the composite ceramic panel was observed by a field emission scanning electron microscope (Carle Zeiss Instruments, GeminiSEM 300 model, Baden-Wurttemberg, Germany). According to GB/T 30873-2014, the electromagnetic heating test of the composite ceramic panel was carried out by an electromagnetic furnace (Haier Instruments, C21-H1202 model, Qingdao, China).

## 4. Results and Discussion

### 4.1. Thermal Stress Analysis of the Composite Panel with a Single Layer Structure and a Gradient Structure Magnetic Layer

Compared with a single-layer coating, a gradient structure can effectively alleviate the stress mutation on the interface between the magnetic layer and ceramic layer. A rational gradient structure of the magnetic layer can improve the internal stress distribution of the coating, and significantly improve the structural stability and service life of the composite ceramic panel [16]. In order to ensure the stability quality of the composite ceramic panels during the electromagnetic heating process and to avoid too large a stress mutation causing the panel to crack, the model of an alumina–magnetic material–zirconia sandwich structure was established by ANSYS software. This model simulated the thermal stress field of the composite panels with a single layer or a three-layer gradient structure. The analysis results of the temperature field of the composite ceramic panel with a single-layer structure and a three-layer gradient structure are shown in Figure 5. As can be seen from Figure 5, compared with the non-gradient structure, the gradient structure had an obvious effect on the temperature characteristics. The peak temperature decreased by 32.6% from 687.982 °C to 463.351 °C. The gradient structure effectively alleviated the stress mutation of the composite ceramic panel during the electromagnetic heating and avoided cracking the panel due to the temperature rising too fast.

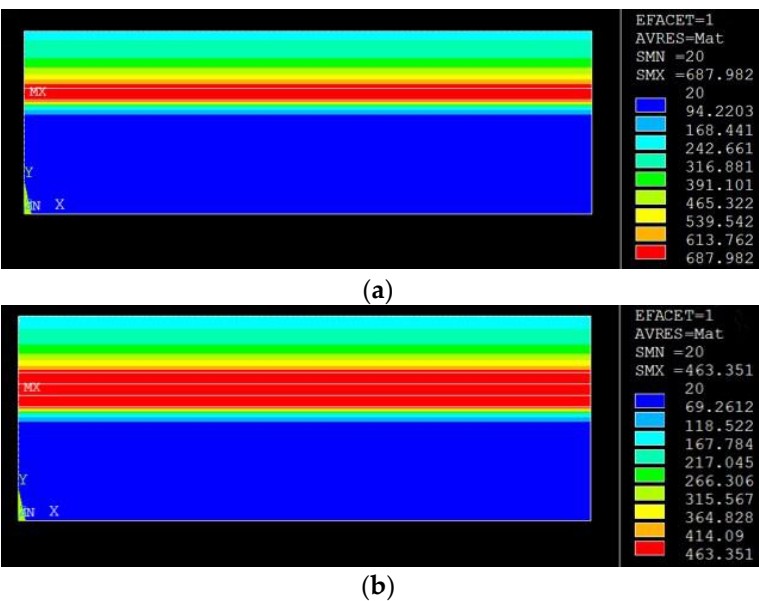

**Figure 5.** Results of the temperature field of the different composite ceramic panels: (**a**) with a single magnetic interlayer; (**b**) with a three-layer gradient.

The more uniform the temperature change during heating, the smaller the likelihood of the ceramic panel cracking due to thermal stress. Figure 6 shows the effect of different magnetic layer structures on the shear stress of the composite ceramic panels. It can be seen that the gradient structure changed the thermal stress distribution pattern of the composite ceramic panel. In the single-layer structure model, there was a 308 MPa stress mutation between the alumina layer and the magnetic layer. However, when the three-layer gradient structure was adopted, the maximum stress occurred at the interface between the alumina layer and the magnetic layer. The stress amplitude decreased by 37.7% from 308 to 192 MPa, which greatly reduced the stress mutation, improved the problem of excessive shear stress caused by temperature change in the magnetic layer, and effectively prevented the cracking of the composite ceramic panel caused by shear stress [13].

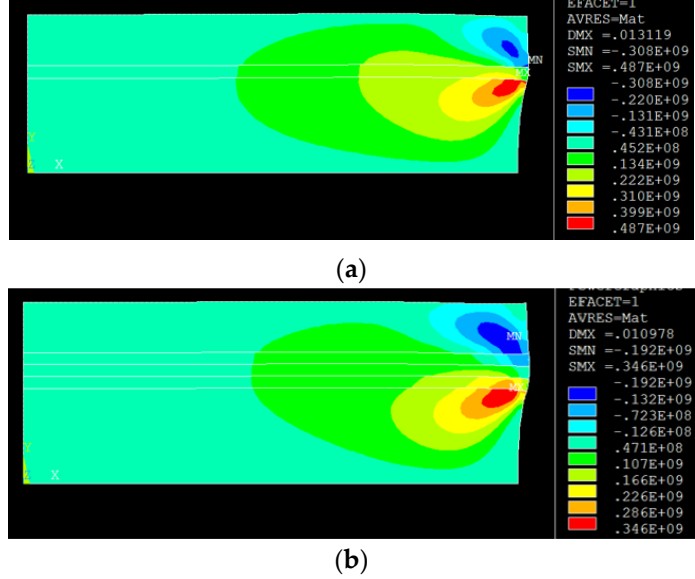

**Figure 6.** Shear stress distribution of the composite ceramic panels: (**a**) with a single magnetic interlayer; (**b**) with a three-layer gradient.

### 4.2. Effect of the Glass Powder Content on the Micromorphology of the Magnetic Layer

As a heating element of a composite ceramic panel, the magnetic layer requires good electromagnetic properties after sintering. Low-melting glass powder is one of the main components of the magnetic layer. The wettability of the low-melting glass powder to the substrate and the metal powder determines the bonding strength of the film layer and the ceramic matrix, and the electrical conductivity of the film layer after sintering [17]. Therefore, the glass powder content has a great influence on the sintering quality of the magnetic layer. In this study, the micromorphology of the alumina panel surface and the cross-section of the composite ceramic panel with the magnetic layer after sintering was characterized by SEM. As can be seen from the surface of the magnetic layer and the cross-section of the composite ceramic panel (Figure 7), the quality of the film first increased and then decreased with an increase in the glass powder content. When the glass powder content reached 9 wt%, the film had the best quality. When the glass powder content was 9 wt%, the surface of sample A3 was dense, and the functional phase particles in the micromorphology were in contact with each other to form a planar-like structure, and the film layer was tightly bonded to the ceramic substrate. However, a large number of defects occurred on the surface and cross-section of samples A1 and A5. The surface of sample A1 was rough, and the copper and iron particles in the film layer were randomly distributed with each other, and the cross-section structure was loose and retained many pores. Since the glass powder content of sample A1 was relatively small, the sintering temperature of the film layer was correspondingly increased, and it was difficult to compactly sinter at 950 °C. Moreover, the content of the liquid glass was too low to effectively drive the creep of metal particles for rearrangement. Although the increase in the glass phase content could improve the wettability of the film layer to the ceramic matrix and enhance the bonding strength between the film layer and the ceramic matrix, the copper-iron particles were correspondingly reduced as the glass powder content increased. In addition, the excess molten glass was wrapped on the surface of the metal particles, so it was difficult for the metal particles to come into direct contact with each other to form a chain-shaped conductive path. As shown in Figure 7e, the local surface of the film layer was entirely composed of glass without copper-iron particles. In addition, the sintering temperature of 950 °C was much higher than the softening temperature (570 °C) of the glass added in this study. A large amount of molten glass boiled on the surface of the film, resulting in the formation of pores on the surface of the film and a decrease in the density. This was consistent with the experimental results in [18].

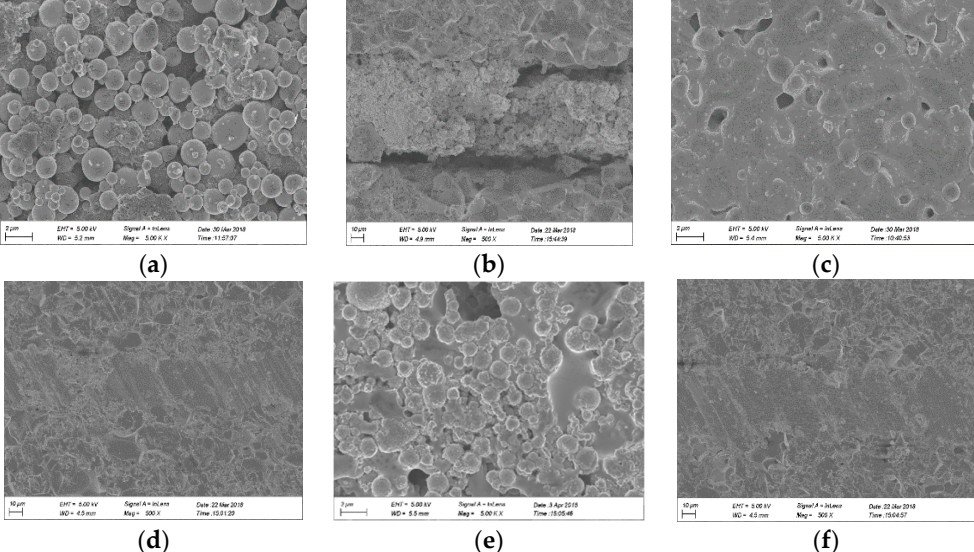

**Figure 7.** Micromorphology of the surface and cross-section of the A1, A3, and A5 coatings: (**a**) surface of A1; (**b**) cross-section of A1; (**c**) surface of A3; (**d**) cross-section of A3; (**e**) surface of A5; and (**f**) cross-section of A5.

### 4.3. Thermal Shock Resistance of a Single-Layer Structure and Three-Layer Gradient Composite Ceramic Panel

According to the results of the finite element simulation analysis, five cycles of electromagnetic heating tests were carried out on the composite ceramic panel with a single-layer structure of A3 slurry with the best film quality and a three-layer gradient structure. One cycle was composed of heating the ceramic panel to 120 °C for 30 min and then cooling it to room temperature naturally. The heating power of the electromagnetic furnace was 900 W. Figure 8 shows the SEM photograph of the surface and cross-section morphologies of the composite ceramic panel after the electromagnetic heating test. It can be seen that the thermal shock resistance of the composite ceramic panel with a three-layer gradient structure was excellent. There were no cracks or spalling on the panel after five cycles of electromagnetic heating. It can be seen from the cross-section of the composite substrate that the middle layer of the composite ceramic panel was compact with no obvious pores or shrinkage, and was closely bound to the ceramic matrix, as shown in Figure 8d. However, the single-layer composite ceramic panel cracked after only one heating cycle, and the cracks were mainly distributed in the alumina ceramic panel. It is worth noting that the cross-section of the non-expanding part of the composite ceramic panel with the single-layer magnetic interlayer was compact and without obvious porosity or shrinkage.

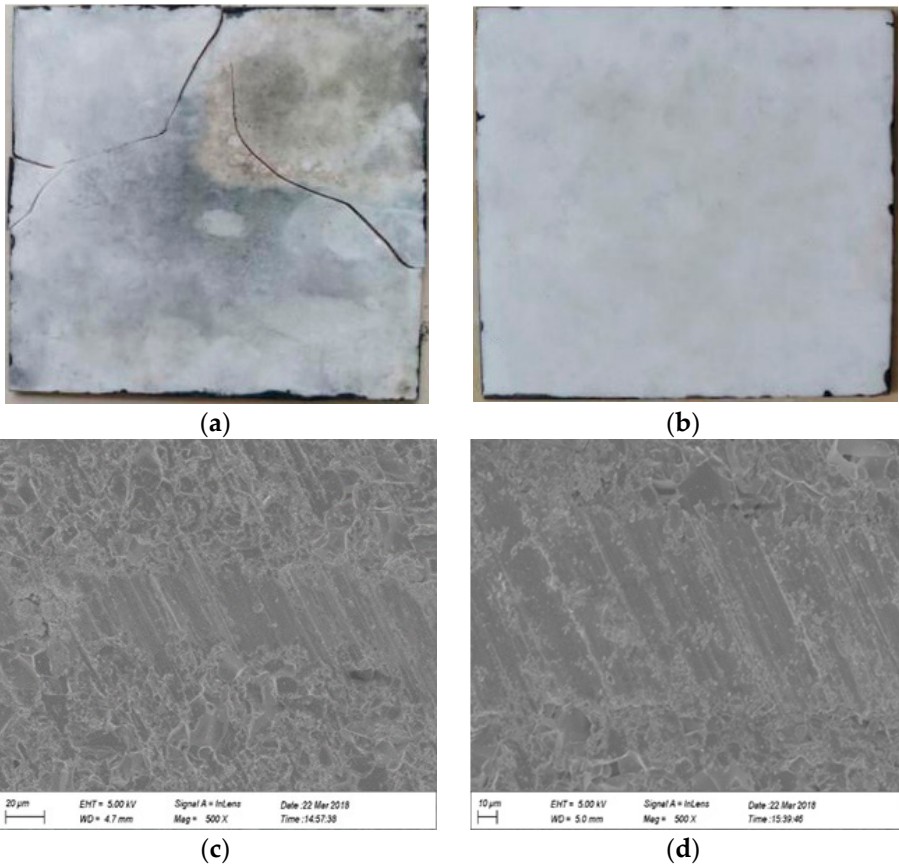

**Figure 8.** Morphology of the composite ceramic panel after the electromagnetic heating test: (**a**) with a single magnetic interlayer; (**b**) with a three-layer gradient; (**c**) cross-section of the panel with a single magnetic interlayer; and (**d**) cross-section of the panel with a three-layer gradient.

The thermal shock resistance of the composite ceramic panel was determined by the thermal expansion coefficient of each layer of material and the strength of the ceramic layer. Tetragonal zirconia ceramics have high strength and good thermal shock resistance, so most of the failure of the composite ceramic panel occurs in the aluminum oxide layer. The closer the thermal expansion coefficient of the magnetic layer is to that of the alumina ceramic matrix, the smaller the probability of cracks

caused by the thermal stress. Since the thermal expansion coefficient of the magnetic interlayer with a single-layer structure is quite different from that of the ceramic substrate (as shown in Tables 4 and 5), a large thermal stress was generated between the magnetic interlayer and the ceramic substrate during electromagnetic heating. When the thermal stress exceeded the tensile strength of the alumina substrate, the magnetic layer was tightly combined with the ceramic substrate due to the presence of the glass phase, and the composite ceramic panel was prone to expansion and cracking. This was consistent with the analysis in [19]. The gradient structure of the magnetic interlayer could reduce the interfacial stress between the magnetic interlayer and the ceramic layer during the electromagnetic heating process and improve the thermal shock resistance of the composite ceramic plate.

## 5. Conclusions

In this study, a kind of alumina–magnetic interlayer–zirconia sandwich structure composite ceramic panel was prepared, and the temperature field and macro-stress field of the composite ceramic panel material during heating were simulated by the finite element model. The magnetic interlayer consisted of a copper phase, iron phase, and glass phase. Based on the results obtained, the following conclusions can be drawn:

1.　When the glass powder content was 9 wt%, the magnetic layer had the best quality, and the film was closely bound to the ceramic substrate.
2.　The magnetic layer with the gradient structure could effectively improve the shear stress mutation caused by the temperature change and so prevented the expansion and cracking of the composite ceramic panel due to excessive stress during the heating process. The three-layer gradient structure could reduce the stress between the aluminum layer and the magnetic layer from 308 to 192 MPa, which significantly improved the thermal shock resistance of the composite ceramic plate. The gradient structure of the magnetic layer could ensure the stability of the composite ceramic panel after five cycles of electromagnetic heating.

**Author Contributions:** S.W. conceived and designed the experiments; A.P., Y.H., and P.C. analyzed the models and data; A.P., H.L., and Y.H. performed the experiments; and S.W., A.P., and H.L. were responsible for the manuscript writing.

**Funding:** This research was funded by the Science and Technology Projects of Guangdong Province, No. 2017B090922008 and the Science and Technology Projects of Chaozhou, No. 2018ZD10.

**Acknowledgments:** The authors would like to thank the South China Institute of Fine Ceramics Technology for providing experimental support.

**Conflicts of Interest:** The authors declare no conflict of interest.

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
