# Peer review of "Study on the Preparation Technology of a Ceramic Panel with a Magnetic Interlayer for an Induction Cooker"

_applsci, doi:10.3390/app9050970_

Round 1

Reviewer 1 Report

The paper presents a numerical study aimed to describe the thermal behaviour of a sandwich ceramic panel. The topic is worth of investigation and it is aligned with the aims and scopes of the journal. The paper is well organized. The introduction and the numerical methodology are a little bit poor. However, the results seem to offer some good insights.

Some general points that need to be addressed by the authors to enhance the quality of the manuscript are reported below:

1.     The Introduction needs to be improved. For example, you might talk about sandwich structures in general and their application in different context.  For example, in civil, aircraft and marine engineering. Since that you perform FE simulation, I strongly suggest to improve the references about other numerical model used in different contexts where sandwich panels are used  (e.g. recently other authors have published some works in which numerical methodologies based on moving mesh approach has been used to simulate static or dynamic crack propagation [DOI: https://doi.org/10.1016/j.compstruct.2018.06.113 and DOI: https://doi.org/10.3221/IGF-ESIS.47.21]) or also other numerical methodologies to simulate the thermal loads (e.g DOI: 10.1080/01495730500360385) and so on…..

2.     Figure 6 is in bad quality. Please replace It

3.     Eq. (3) and (4) are in bad quality. Please replace them.

4.     Please provide more information about the numerical strategy adopted to solve the numerical problem (i.e. Explicit or implicit algorithm?). I strongly suggest the authors to explain the reasons about the choice of a method rather that another one. Please, I would suggest the authors to provide also some examples in the framework of thermal analysis and but also in other engineering issues (e.g. explicit algorithm “ Shear performance of FRCM strengthened RC beams. Paper presented at the American Concrete Institute, ACI” or implicit https://doi.org/10.1002/nme.1620350404 and https://doi.org/10.1016/j.compositesb.2014.12.014)

5.     In the row 113 the authors say: “In order to simplify the calculation, the effect of temperature on the performance parameters of the alumina and zirconia substrates during electromagnetic heating was neglected”. Please, provide more details about this modelling choice.

6.     I strongly reccomend to report the outline of the paper at the end of the introduction.

Overall is a good paper, deserving to be considered for publication in this journal. I advise the authors to revise the paper based on the comments given.

Author Response

Point 1: The Introduction needs to be improved. For example, you might talk about sandwich structures in general and their application in different context.  For example, in civil, aircraft and marine engineering. Since that you perform FE simulation, I strongly suggest to improve the references about other numerical model used in different contexts where sandwich panels are used.

Response 1: Thanks for your advices. According to the reviewer’s advice, we have added the statement on the sandwich structures and FE simulation in the Introduction (line 61-71). “The sandwich structure, which consists of two external high density materials and a thick core made from low density material, has been widely applied in aerospace, aircraft, marine industry and civil engineering. The configuration coupled with an optimized material combination significantly improves the overall performance of the system and structure as well as the efficiency of material use. However, the working environment of high temperature and impact load will cause problems such as cracks in the sandwich structure. Based on moving mesh approach, Marco et al.[10] proposed a numerical method methodology to investigate the behavior of composite sandwich structures under static and dynamic loading conditions. It could correctly simulate interfacial crack onset, layer kinematic and debonding propagation. Based on 3D FEM, Xue et al.[11] simulated the thermal-structrual response of complicated sandwich composites. FEM has been one of the most effective soft to assist manufacture.”

Point 2: Figure 6 is in bad quality. Please replace It.

Response 2: Thanks for your suggestion. According to the reviewer’s advice, we have replaced Figure 6 with a picture in good quality ”

Point 3: Eq. (3) and (4) are in bad quality. Please replace them.

Response 3: Thanks for your suggestion. we have replaced Eq. (3) and (4).

Point 4: Please provide more information about the numerical strategy adopted to solve the numerical problem. (i.e. Explicit or implicit algorithm?). I strongly suggest the authors to explain the reasons about the choice of a method rather that another one.

Response 4: Thanks for your suggestion.

(1)   According to the reviewer’s advice, we have added the numerical strategy adopted to solve the numerical problem(line 115-117). “By dividing the continuum into a limited number of elements, then combining these elements into a whole, introducing boundary conditions and establishing algebraic equations, the displacement and stress of the continuum at discrete points are finally obtained.”

(2)   The main research is about stress and strain, so we used the implicit finite element method.

Point 5: In the row 113 the authors say: “In order to simplify the calculation, the effect of temperature on the performance parameters of the alumina and zirconia substrates during electromagnetic heating was neglected”. Please, provide more details about this modelling choice.

Response 5: Thanks for your suggestion. In this finite element analysis, the properties of alumina or zirconia substrates, such as hardness or coefficient of thermal expansion, are relatively stable under the low cooking temperature, so the effect of temperature was neglected.

Point 6: I strongly reccomend to report the outline of the paper at the end of the introduction.

Response 6: Thanks for your suggestion. According to the reviewer’s advice, we have modified the paragraph at the end of the introduction as following (line 72-81).  “In this paper, the composite ceramic plate with sandwich structure is prepared. In section 2, the finite element model of alumina–magnetic material–zirconia (in a sandwich structure) composite ceramic panel under thermal load was established by ANSYS software. The temperature field and the macroscopic stress field of the composite ceramic panel were simulated. In section 3, based on the results of finite element analysis, a magnetic slurry was developed and screen printed onto the tape casted Al2O3 and ZrO2 ceramic panel. The composite ceramic panel with gradient magnetic interlayer was sintered at 950 °C under protective atmosphere. Finally, in section 4, the influence of the magnetic layer structure on the thermal stress distribution of the composite ceramic panel was analyzed. The micromorphology and thermal shock resistance behavior of the panel under different condition were investigated.”

Reviewer 2 Report

Dear Authors, 

I have read your manuscript carefully and I would say that this manuscript would be very interesting for readers. The objectives of the study are clearly defined. The introduction provides a good, generalized background of the topic. The results are clearly explained and are presented in an appropriate format. The figures and tables show essential data; some of the data are also summarized in the text. I do not think any additional graphics are necessary. The cited literature is relevant to the study and balanced. This manuscript was well prepared and could be published after correcting some editing errors. 

When presenting results in tables, the authors should always use identical figures, e.g. if in one case the authors give the result of density measurement with an accuracy of two decimal places, they should give it all the time. Currently, the authors write 3.97 once, and another time 6, and it should be 6.00. This remark applies to the presentation of all results in the manuscript.

Line 149: The name of SEM is written using various fonts.

Figure 7 presents the morphology of the surface, not microstructure.

I would suggest using the abbreviation “min” instead of “minutes”.

Author Response

Point 1: When presenting results in tables, the authors should always use identical figures.

Response 1: Thanks for your advices. According to the reviewer’s advice, we have unified the measurement accuracy.

Point 2: Line 149: The name of SEM is written using various fonts.

Response 2: Thanks for your suggestion. We have checked and revised our manuscript carefully.

Point 3: Figure 7 presents the morphology of the surface, not microstructure.

Response 3: Thanks for your advices. We have revised the description in our manuscript carefully.

Point 4: I would suggest using the abbreviation “min” instead of “minutes”.

Response 4: Thanks for your suggestion. We have used the abbreviation “min” instead of “minutes” throughout this article.

Reviewer 3 Report

This paper describes the production method and structural features of a layered material being developed for the use in induction cookers. The authors suggest a gradient structure of the magnetic interlayer as a way to reduce stresses between the ceramic layer and the magnetic metal-containing layer. A concentration of a glass additive to the magnetic layer was found ensuring good bonding of the layer to ceramic plates. I think this paper deserves publication in the Journal after minor revision.

In the Abstract, please provide information on the chemical and phase composition of ceramic and magnetic layers. This will make it clearer for the reader what material you have developed.

Also, please provide information on the acceptable thickness range of the layers in the Abstract.

For a broad audience that will read your article, please comment on what "umami" taste is.

Could you provide references (patents, papers, internet resources) for these statements: "In 2015, Tiger introduced a full-clay electromagnetic rice cooker, which was formed by spraying a layer of a magnetic heater on the bottom of the full-clay earth cooker and firing it at high temperature [??]. By embedding a ferromagnetic plate in the bottom of a sintered ceramic pot, Chunshui Chen succeeded in heating a ceramic pot by electromagnetic induction [??]".

Tables should be re-numbered. Table 1 is mentioned in the text in Section 3. Other tables are mentioned earlier in the text but are numbered Table 2, 3, etc.

Please add information on the composition of the glass powder and organic binder that you used (to Section 3). What is meant by additive in Table 1?

Information on the particle size of the powders used in this work is missing.

Sometimes, Conclusions are read separately from the rest of the article. So, please provide the material composition information in the Conclusion Section as well.

Author Response

Point 1: In the Abstract, please provide information on the chemical and phase composition of ceramic and magnetic layers.

Response 1: Thanks for your advices. According to the reviewer’s advice, we have added the information on the chemical and phase composition of ceramic and magnetic layers in the Abstract (line 10-13 and line 14-16). “a kind of sandwich structural composite ceramic panel, which consists of Al2O3 ceramic panel, magnetic heating interlayer and ZrO2 ceramic substrate, was developed by combining the tape casting process and the screen printing process.” “In this research, the slurry composition of functional phase, glass powder and organic carrier was optimized for preparing the heating interlayer with excellent electromagnetic properties. ”

Point 2: Also, please provide information on the acceptable thickness range of the layers in the Abstract.

Response 2: Thanks for your suggestion. According to the reviewer’s advice, we have added the information on the acceptable thickness range of the layers in the Abstract (line 19-20). “In the range of 0.1-0.3 mm thickness of magnetic heating layer, the temperature field and the macroscopic stress field of the composite ceramic panel were simulated.”

Point 3: For a broad audience that will read your article, please comment on what "umami" taste is.

Response 3: Thanks for your suggestion. Humans enjoy five kinds of taste buds (possibly six): sour, bitter, salty, umami (or meatiness) and sweet (as well as possibly fat). We have modified this statement as the following (line 47). “it not only can improve the heating efficiency of the ceramic pot, but also can maintain the taste buds of the food itself.”

Point 4: Could you provide references (patents, papers, internet resources) for these statements: "In 2015, Tiger introduced a full-clay electromagnetic rice cooker, which was formed by spraying a layer of a magnetic heater on the bottom of the full-clay earth cooker and firing it at high temperature [??]. By embedding a ferromagnetic plate in the bottom of a sintered ceramic pot, Chunshui Chen succeeded in heating a ceramic pot by electromagnetic induction [??]".

Response 4: Thanks for your suggestion. According to the reviewer’s advice, we have respectively  added the reference[8] and [9] at line 54 and line 55.

Point 5: Tables should be re-numbered. Table 1 is mentioned in the text in Section 3. Other tables are mentioned earlier in the text but are numbered Table 2, 3, etc.

Response 5: Thanks for your suggestion. According to the reviewer’s advice, we have re-numbered  all tables.

Point 6: Please add information on the composition of the glass powder and organic binder that you used (to Section 3). What is meant by additive in Table 1?

Response 6: Thanks for your suggestion. According to the reviewer’s advice, we have added the information on the composition of the glass powder and organic binder (line 166 and line 167).  And additive is used as surfactants and thixotropic agents.  Since the organic carrier component has been added, in order to avoid misunderstanding, we have revised the table and classified its content into the organic carrier component.

Point 7: Information on the particle size of the powders used in this work is missing.

Response 7: Thanks for your suggestion. According to the reviewer’s advice, we have added the information on the particle size of the powders used in this work (line 146-147). “In Table 5, the particle median diameter of copper powder, iron powder and glass powder are 1 μm, 1 μm and 3 μm, respectively.”

Point 8: Sometimes, Conclusions are read separately from the rest of the article. So, please provide the material composition information in the Conclusion Section as well.

Response 8: Thanks for your suggestion. According to the reviewer’s advice, we have added some sentences about the material composition in the Conclusion Section (line 287-291). “In this study, a kind of alumina-magnetic interlayer-zirconia sandwich structure composite ceramic panel was prepared, and the temperature field and macro-stress field of composite ceramic panel material during heating were simulated by finite element model. The magnetic interlayer is consisted of copper phase, iron phase and glass phase.  Based on the results obtained, the following conclusions can be drawn.”
